# M(II)Al_4_ Type Layered Double Hydroxides—Preparation Using Mechanochemical Route, Structural Characterization and Catalytic Application

**DOI:** 10.3390/ma14174880

**Published:** 2021-08-27

**Authors:** Márton Szabados, Adél Anna Ádám, Zsolt Kása, Kornélia Baán, Róbert Mucsi, András Sápi, Zoltán Kónya, Ákos Kukovecz, Pál Sipos

**Affiliations:** 1Department of Organic Chemistry, University of Szeged, Dóm tér 8, H-6720 Szeged, Hungary; szabados.marton@chem.u-szeged.hu; 2Material and Solution Structure Research Group, Institute of Chemistry, University of Szeged, Aradi Vértanúk tere 1, H-6720 Szeged, Hungary; adeladam@chem.u-szeged.hu (A.A.Á.); kasa.zsolt@chem.u-szeged.hu (Z.K.); 3Department of Applied and Environmental Chemistry, University of Szeged, Rerrich B. tér 1, H-6720 Szeged, Hungary; kornelia.baan@chem.u-szeged.hu (K.B.); muule93@gmail.com (R.M.); sapia@chem.u-szeged.hu (A.S.); konya@chem.u-szeged.hu (Z.K.); kakos@chem.u-szeged.hu (Á.K.); 4MTA-SZTE Reaction Kinetics and Surface Chemistry Research Group, Rerrich B tér 1, H-6720 Szeged, Hungary; 5Department of Inorganic and Analytical Chemistry, University of Szeged, Dóm tér 7, H-6720 Szeged, Hungary

**Keywords:** gibbsite intercalation, copper- and magnesium-poor layered double hydroxides, mechanochemistry, catalytic carbon monoxide oxidation

## Abstract

The synthesis of the copper-poor and aluminum-rich layered double hydroxides (LDHs) of the CuAl_4_ type was optimized in detail in this work, by applying an intense mechanochemical treatment to activate the gibbsite starting reagent. The phase-pure forms of these LDHs were prepared for the first time; using copper nitrate and perchlorate salts during the syntheses turned out to be the key to avoiding the formation of copper hydroxide sideproducts. Based on the use of the optimized syntheses parameters, the preparation of layered triple and multiple hydroxides was also attempted using Ni(II), Co(II), Zn(II) and even Mg(II) ions. These studies let us identify the relative positions of the incorporating cations in the well-known selectivity series as Ni^2+^ >> Cu^2+^ >> Zn^2+^ > Co^2+^ >> Mg^2+^. The solids formed were characterized by using powder X-ray diffractometry, UV–Vis diffuse reflectance spectroscopy, Fourier-transform infrared spectroscopy, thermogravimetric analysis and scanning electron microscopy. The catalytic potential of the samples was investigated in carbon monoxide oxidation reactions at atmospheric pressure, supported by an in situ diffuse reflectance infrared spectroscopy probe. All solids proved to be active and the combination of the nickel and cobalt incorporation (which resulted in a NiCoAl_8_ layered triple hydroxide) brought outstanding benefits regarding low-temperature oxidation and increased carbon monoxide conversion values.

## 1. Introduction

Layered double hydroxides (LDH) belong to a group of anionic clay-type materials possessing a CdI_2_-type structure. Due to the chemical diversity offered by variability in the composition of the interlamellar galleries and layers, they can be finely tuned to numerous applications, from healthcare [1] to agriculture [2] and the polymer industry [3], just to mention a few. Perhaps the most promising research field of the pristine and heat-treated derivatives of LDHs is their application as catalysts. The majority of LDHs have an intrinsically basic character and the cationic components, with alternating oxidation states, serve as redox centers. Hence, these compounds are commonly employed in transesterification [4], Michael addition [5], or even in photochemical transformations [6]. The following formula is used for the description of the LDHs: [M^2+^_1−x_M^3+^_x_(OH)_2_]^x+^[A^n−^_x/n_·mH_2_O)]^x−^, where M^2+^ and M^3+^ are the di- and trivalent metal ions, respectively, while A^n−^ stands for the interlamellar anions with a charge of *n*, and *x* = M^3+^/[M^2+^ + M^3+^] [7]. The formation of their framework can be deduced in two ways, depending on the value of *x*. When *x* ≤ 0.5, the LDHs can formally be derived from the brucite-like (Mg(OH)_2_) layered divalent metal hydroxides (with M^2+^ = Mg^2+^, Ca^2+^, Ni^2+^, Co^2+^, Cu^2+^, Zn^2+^, etc.) via the isomorphous substitution of divalent cations with trivalent ones (e.g., Al^3+^, Cr^3+^, Fe^3+^, etc.). Thus, in the layers, a positive charge is built up and (hydrated) anions intercalate to compensate for the charge of the layer. The preference of intercalating various inorganic anions follows the lyotropic series of CO_3_^2−^ >> SO_4_^2−^ >> OH^−^ > F^−^ > Cl^−^ > Br^−^ > NO_3_^−^ > I^−^ ≈ ClO_4_^−^, which is mainly controlled by the size and the strength of the electrostatic interaction [8,9].

At higher *x* values, the incorporation of divalent ions (or, e.g., the monovalent Li^+^) also results in the formation of an LDH-type structure; however, so far, only aluminum trihydroxide was found to be suitable for generating these M(III)-rich and, thus, M(I)-, M(II)-poor layered double hydroxides (with the following general formula: [M(I)_2_/M(II) − M(III)_4_(OH)_12_]^2+^[A^n−^_2/n_·mH_2_O)]^2−^). The key is the structural similarity of their frameworks to that of brucite, but appearing here with vacant octahedral holes in which the extraneous metal cations could incorporate, resulting in positively charged layers. This creates the need for the formation of the anion-containing interlamellar galleries. These LDHs have excellent ion-exchange properties [10] and modified anion exchange capability, with a preference for the tetrahedral anions and, hence, a lower affinity for the CO_3_^2−^ anions compared to that of the more conventional (*x* ≤ 0.5) LDHs [11,12].

The success of the incorporation of extraneous metal ions and the generation of the LDH phase is limited by a variety of factors. These are the ionic radii and the solvation energies of the metal cations [11], the solubility of the metal salts used, and the quality of the interlamellar anions with regard to their position in the lyotropic series. Therefore, the incorporation of the Li^+^ (having a suitable ionic radius, good solubility in water, and relatively low hydration enthalpy), and the formation of LiAl_2_–LDH in the direct reaction of the aluminum hydroxide and various lithium salts resulted in the occupation of all the vacant holes and the intercalation of numerous inorganic anions: X^−^ = Cl^−^, Br^−^, NO_3_^−^, SO_4_^2−^, OH^−^ and CO_3_^2−^ [13,14]. For the incorporation of M(II) cations, several studies were reported in the last two decades; however, until now, the successful incorporation of Mg^2+^, Co^2+^, Ni^2+^, Cu^2+^ and Zn^2+^ meant the occupation of half of the octahedral holes, resulting in M(II)Al_4_–LDHs. For this, the use of harsh synthesis conditions, that is, concentrated (2–10 M) metal NO_3_^−^, SO_4_^2−^ and Cl^−^ salt solutions, hydrothermal conditions (120–150 °C reaction temperature), and long reaction times (24–72 h) were necessary [15,16,17,18]. Moreover, so far, the synthesis of the copper-containing variants in a phase-pure form was found to be impossible, due to the concomitant formation of copper hydroxide impurities. Hence, the position of the Cu(II) in the selectivity series of cation incorporation (Li^+^ >> Ni^2+^ >> Co^2+^ ≈ Zn^2+^) was unknown [19]. The case of MgAl_4_–LDHs is somewhat similar, in the sense that preparation was found to be rather complicated [16,17].

In the synthesis procedures, the application of mechanochemical pretreatment is not a necessity [11,12,13,14,16,18,20]; however, in several cases, mechanically activated aluminum hydroxide particles were used for obtaining these LDHs [10,15,17,19,21]. Recently, a large number of simple-to-set-up, moderately priced instruments with specific operating characteristics have become commercially available, making it possible to carry out various mechanochemical treatments for the preparation of well-known [22] as well as unique LDHs [23]. In our recent work, it was demonstrated that high-energy milling could induce a unique polymorph transformation of Al(OH)_3_ and facilitate the generation of LDHs [24].

Therefore, our aim was to explore the possibility of synthesizing phase-pure copper-containing aluminum-rich layered double hydroxides of a CuAl_4_ type. Based on the knowledge gained, the same type of synthesis was attempted, using multiple metal ions (the parallel incorporation of Mg(II), Ni(II), Co(II) and Zn(II) ions to gain layered triple and multiple hydroxides) using intense mechanochemical pretreatment. The influence of the counter anions of the starting copper salts on the synthesis and catalytic activities of the as-prepared materials were also explored in catalytic carbon monoxide oxidation.

## 2. Materials and Methods

### 2.1. Materials

The nickel, copper and magnesium salts (Ni(NO_3_)_2_·6H_2_O, Ni(ClO_4_)_2_·6H_2_O, Cu(NO_3_)_2_·3H_2_O, CuCl_2_·2H_2_O, CuBr_2_, Cu(ClO_4_)_2_·6H_2_O, Mg(NO_3_)_2_·6H_2_O, and Mg(ClO_4_)_2_·6H_2_O) were purchased from Merck (St. Louis, MO, USA), while the zinc and cobalt salts (Zn(NO_3_)_2_·6H_2_O, Zn(ClO_4_)_2_·6H_2_O, Co(NO_3_)_2_·6H_2_O, Co(ClO_4_)_2_·6H_2_O) were provided by the Alfa Aesar (Kandel, Germany). Anhydrous Al(OH)_3_ (gibbsite) was received from Reanal Private (Budapest, Hungary). The chemicals were of 99%+ purity and were used without additional purification.

### 2.2. Mechanical Pretreatment of Gibbsite

For the pre-milling steps, a Retsch MM 400 mixer mill (Retsch GmbH, Haan, Germany), equipped with two stainless steel grinding balls of ~8.2 cm^3^ (25 mm diameter) and grinding jars (50 cm^3^ inner volume), were used. In all cases, the dry-milling of the Al(OH)_3_ was executed at 12 Hz grinding frequency and 100 ball/sample mass ratio. (We would like to highlight here that the specific structural changes caused by a vibration/mixer mill are in part different from those that evolve in a rolling or planetary ball mill during its circular motion [25]; here, the jars pass in radial oscillations along the horizontal axis and, inside, the balls collide with the rounded ends of the jars, resulting in intense collisions of short duration.)

### 2.3. Preparation of the Magnesium-, Copper-, Nickel-, Cobalt- and Zinc-Poor M(II)Al_4_-LDHs

The first step of each synthesis was the mechanochemical pretreatment of aluminum trihydroxide. In our recent work, it was shown that 4 h of grinding time and 3 days of impregnation were the optimal conditions for the preparation of phase-pure NiAl_4_–LDHs [24]. In this study, the preparation of cobalt, copper, zinc and magnesium salts required an increase of milling duration of up to 6–12 h, and 4 days of impregnation time. In all cases, 100 mg of activated Al(OH)_3_ solid was placed in a glass tube with the metal salt dissolved in 5 cm^3^ distilled water. The initial M(II):Al ratio was systematically varied in a wide range to test its influence on the success of LDH preparation. The synthesis of the layered triple (LTHs) and multiple hydroxides (LMHs) was executed by the addition of a mixture of the required metal salts to the gibbsite powder. The obtained suspensions were stirred at 90 °C with a 1000 rpm stirring rate under an air atmosphere. The samples were collected on filters with 0.45 μm pore size, washed with distilled water (200–1000 mL) several times (the protocol depended on the concentration of the impregnating solution), and were dried at 80 °C overnight.

### 2.4. Carbon Monoxide Oxidation in a Continuous Flow Reactor

The catalytic reactions were performed in a fixed-bed continuous-flow reactor (University of Szeged, Szeged, Hungary) (8 mm internal diameter, 200 mm length), externally heated with a thermocouple between 100 and 700 °C; 100 ± 2 mg samples were weighed and placed between quartz wool plugs. At atmospheric pressure, discontinuous heating was used and the reactor was heated up at a 10 °C/min rate; the targeted temperatures were held for 25 min, and the gas samplings were carried out in the seventh minute. Prior to the measurements being taken, the solids were degassed at 110 °C for 2 h in an Ar atmosphere. The flow rate and composition of the reacting gas mixture were as follows: carbon monoxide—4 cm^3^/min, oxygen—10 cm^3^/min, and argon—46 cm^3^/min. To avoid any condensation, the tubes delivering the gases were also externally heated between the reactor and the HP 5890 gas chromatograph (Hewlett-Packard, Waldbronn, Germany) equipped with a Porapak Q packed column for thermal conductivity detection. The analysis of the composition of the products was performed using 120 °C injections and a 250 °C detector temperature; the oven was heated from 45 °C (with a 6-min hold) up to 180 °C, with a 4-min hold and 10 °C/min heating rate.

### 2.5. Methods of Structural Characterization

The main method used to monitor the success of the synthesis and the obtained phase compositions was powder X-ray diffractometry, using a Rigaku Miniflex II diffractometer (Rigaku Corporation, Tokyo, Japan), equipped with a scintillation detector and a Ni foil K*β* filter (operating at 30 kV and 15 mA). The normalized XRD patterns were registered in the θ = 5–80° range using CuKα (λ = 1.5418 Å) radiation, with a 4°/min scan speed in continuous mode and a step width of 0.02° 2θ. The average crystallite sizes of LDHs (coherently scattering domain sizes—the crystallite thicknesses from layers connected to each other in the *c*-axis direction of the platelets) were estimated from the full width at half-maximum (FWHM) of the first reflections, applying Gaussian distribution and the Scherrer equation with a shape factor of 0.9. The reflections of the patterns were identified with the help of the JCPDS-ICDD (Joint Committee of Powder Diffraction Standards—International Centre for Diffraction Data) database.

To investigate the structural properties of the LDHs, Fourier-transform infrared (FT-IR) spectra were registered on a JASCO FT/IR-4700 spectrophotometer (Kyoto, Japan), accumulating 256 scans at 4 cm^−1^ resolution with a ZnSe ATR attachment and DTGS detector. The normalized spectra were obtained in the 4000–600 cm^−1^ wavenumber range.

The metal content of the samples was analyzed by an Agilent 7900 ICP-MS (Agilent Technologies, Santa Carla, CA, USA) (inductively coupled plasma mass spectrometry) spectrometer using ICP multielement standard solution IV (CertiPUR). The solids were dissolved in hydrochloric acid with the aid of microwave digestion.

The optical properties of the LDHs were probed by diffuse reflectance spectroscopy (DRS) with an Ocean Optics USB4000 spectrometer (Ocean Insight, Duiven, Netherlands) and DH-2000-BAL light source. As the white reference, BaSO_4_ was used in the 225–890 nm wavelength range. For the transformation of the reflectance spectra into absorption, the Schuster–Kubelka–Munk function was used, while the optical energy gaps were determined by the extrapolation of the straight section of the modified Schuster–Kubelka–Munk function, as plotted vs. the energy of the incident light (Tauc-plot).

To characterize the thermal behavior of the materials, thermogravimetric and differential thermal analysis (TGA and DTA) were performed in the 40–800 °C temperature range by applying a Netzsch STA 409 PC Luxx derivatograph (Netzsch Holding, Selb, Germany), using a constant flow (60 mL/min) of synthetic air and a 10 °C/min heating rate. The samples were taken into high-purity alpha-alumina crucibles, and 40–50 mg solids were employed.

The morphology of the materials was visualized by a Hitachi S-4700 scanning electron microscope (SEM) (Hitachi Ltd, Tokyo, Japan) at various acceleration voltages. Onto the surface of solids, a few nanometers of a conductive gold film were sublimed to avoid the electrostatic charging of the particles. Elemental analysis was carried out by energy-dispersive X-ray spectroscopy (Roentec AG, Berlin, Germany) measurements (EDXS, Röntec QX2 spectrometer installed with a Be window and coupled to the SEM).

The in situ diffuse reflectance infrared Fourier-transform spectroscopy (DRIFTS) measurements were executed using an Agilent Cary-670 FTIR spectrometer (Agilent Technologies, Santa Carla, CA, USA) equipped with a Harrick Praying Mantis diffuse reflectance cell with two BaF_2_ windows. The samples were warmed linearly up to 550 °C with a 20 °C/min heating rate. The spectra were registered in the 4000–800 cm^−1^ wavenumber range, accumulating 32 scans at 2 cm^−1^ resolution. The purging gas was He and the mixture of the CO and He gas (10 and 90 vol %, 40 cm^3^/min) was introduced in the cell. The spectra of the pristine catalysts were used as background.

## 3. Results and Discussion

### 3.1. Optimization of the Synthesis Parameters for the Preparation of CuAl_4_–LDHs

First, the quality of the copper salt starting reagents was systematically varied, in order to investigate their influence on the success of the preparation of CuAl_4_–X^n−^–LDHs (X^n^^−^ is the anion of the copper salt). The use of all salts resulted in the formation of the corresponding LDHs (the identification of the reflections was carried out according to the literature [16,19]). The variation of the basal spacing (that is, the sum of the thickness of a layer, plus the interlayer distance) was found to depend on the size of the interlamellar anion, X^n^^−^. The smallest size was obtained for CuAl_4_–Cl^−^–LDH (7.5 Å, Appendix A; hereafter, “S” refers to information found in the Supporting Information), while the largest was observed for CuAl_4_–ClO_4_^−^–LDH (9.0 Å, Figure 1). When chloride and bromide salts were used, the formation of various side products was observed; next to the CuAl_4_–Cl^−^–LDHs, the reflections of gibbsite (designated as γ-Al(OH)_3_—JCPDS#70-2038) and the tribasic copper chloride (Cu_2_(OH)_3_Cl, JCPDS#25-1427) were permanently present (Appendix A). For the CuAl_4_–Br^−^–LDHs, the formation of dicopper bromide trihydroxide (JCPDS#74-1652) and the presence of an aluminum oxide bromide (Al_3_O_4_Br, JCPDS#17-0695) phase was registered (Appendix A). For both LDHs, the amount of the side products could be decreased significantly by using a minimal excess of the copper salt reagents (Cu:Al molar ratio of 3.0–3.5).

The preparation of nitrate- and perchlorate-containing LDHs was more favored; applying a suitable amount of the starting reagents (1:1 Cu:Al molar ratio for nitrate, and 2:1 Cu:Al molar ratio for perchlorate salts) resulted in the complete disappearance of the gibbsite phase and the formation of CuAl_4_–NO_3_^−^-LDH and CuAl_4_–ClO_4_^−^–LDH, respectively (Figure 1). There were no reflections corresponding to possible impurities and the elemental analysis revealed, commonly, molar ratios close to 1:4 of Cu:Al, indicating that the amorphous gibbsite starting reagent was present only in minute amounts. Using the optimal synthesis conditions, the baseline shift (stemming partially from the disorders in the amorphous phases) could be decreased significantly (from 40–60 to 4–7 wt % amorphous phase content). The increase of the initial Cu:Al molar ratio facilitated the formation of thicker LDHs; the intensity of the reflections increased and split at around 20 and 22° 2θ for the CuAl_4_–ClO_4_^−^–LDH, which is a common phenomenon. This effect was already observed for LiAl_2_– [26], ZnAl_4_–LDHs [27] and, in our recent work, for NiAl_4_–ClO_4_^−^–LDH and for NiAl_4_–NH_2_SO_3_^−^–LDH [24].

The FT-IR of the solids also verified the formation of phase-pure CuAl_4_–X^n^^−^–LDHs. All the vibration bands observed could be clearly attributed to the characteristic vibrations of the aluminum-rich layered double hydroxides (Appendix A). The broad bands around 3500 cm^−1^ can be connected to the stretching vibration of the network of OH groups in the hydroxide layers. The weak band at 1630 cm^−1^ corresponds to the bending vibration of the interlamellar water molecules. Finally, the deformation and translation modes of the Al−OH moieties were registered under 1000 cm^−1^ [28,29]. The weak *v*_1_ (1050 cm^−1^) and *v*_2_ (820 cm^−1^), and the strong *v*_3_ vibrations (1350 cm^−1^) were recorded for the nitrate interlamellar molecules, and the *v*_3_ and *v_4_* mode of the perchlorate anions could readily be observed at 1085 and 620 cm^−1^, respectively, for the corresponding LDHs [30].

### 3.2. Preparation of Layered Triple- and Multiple Hydroxide Systems

The synthesis of the phase-pure CuAl_4_–X^n^^−^–LDHs guided us to carry on the research started by Williams et al., in which the co-incorporation of M(II) ions was attempted, with the aim of establishing a preference order for the building of various metal ions into the gibbsite structure [19]. In our case, first, the NiCuAl–NO_3_^−^–LTHs were studied in detail. It can be seen from the data shown in Table 1 that the ratio of the incorporated metal ions can be fine-tuned by varying the initial molar ratio of the starting nickel and copper nitrate salts. However, the variation of the added amounts of the nickel salt proved to be beneficial, in the sense that the decreasing Ni:Cu molar ratio did not result in the appearance of the reflections that are characteristic of dicopper nitrate trihydroxide (JCPDS#75-1779) impurities (Appendix A). The SEM-EDXS analysis (not shown) proved that the two cations are not segregated, verifying that these are present as a molecular level mixture—similar observations were made by Williams et al. [19]. The XRD patterns of the LTHs and their basal spacing values were practically identical to those obtained for the CuAl_4_–NO_3_^−^–LDHs. Interestingly, a clear correlation was observed between the decrease in the initial nickel content and the crystallite thicknesses, while the intensity of the second (002l) reflections remained largely unaltered (Appendix A).

When working with cobalt, zinc and magnesium ions, it was necessary to know the optimal synthesis parameters leading to phase-pure LDHs. As expected, these cations are found at the end of the preference order, and the preparation of CoAl_4_– and ZnAl_4_–LDHs required large excess of the nitrate salt reagents; that is, 4:1 initial Co/Zn:Al molar ratios (Appendix A). For MgAl_4_–NO_3_^−^–LDH synthesis, as was more or less expected, the use of an even larger (Mg:Al = 32:1) molar ratio and an increase in the pre-milling time proved to be necessary (Appendix A). Using lower concentrations of the above M(II) ions, the conversion of the starting gibbsite into its dehydrated (boehmite, AlO(OH) JCPDS#83-2384) and polymorph (bayerite, as α-Al(OH)_3_ JCPDS#83-2256) forms was recorded. Based on these observations, for the synthesis of the CoZnAl–, MgCoAl– and MgZnAl–LTHs and the MgCoZnAl–LMH, 4:4:1 and 4:4:4:1 initial molar ratios were employed, respectively, to prepare phase-pure LTH and LMH systems.

In X-ray diffraction, no significant differences between the diffractograms of the LDHs, LTHs, and even LMHs (not shown) were found; however, the ICP analysis revealed large variations in their compositions (Table 1). The incorporation of the Ni^2+^ was about five times more favored than that of the Cu^2+^, and more than 17 and 21 times compared to the Zn^2+^ and Co^2+^, respectively. The same numbers for the incorporation of Cu^2+^, relative to that of the Zn^2+^ and Co^2+^, were 7 and 9, respectively. The ratio between the incorporated Co^2+^ and Zn^2+^ was about 1.5 when only these two metals were present during the building into the gibbsite framework. Definitely, the incorporation of the Mg^2+^ ions was the most negligible, even compared to the Zn^2+^ and Co^2+^; its amount was 20–50 times lower in the corresponding LTHs. These differences changed slightly for the LMHs due to the simultaneous and competing incorporation of more than two metal cations. These experiments were repeated in part, using perchlorate salts as starting materials. The affinity order found for perchlorate-containing systems were largely identical with those found for nitrate-containing ones (Appendix A). These data led us to expand the well-known selectivity series [19] as follows:Li^+^ >> Ni^2+^ >> Cu^2+^ >> Zn^2+^ > Co^2+^ >> Mg^2+^

As has been discussed elsewhere, the incorporation of the extraneous cations is mainly determined by the ionic radii and the solvation enthalpies of the metal ions [11,19]. However, the Shannon–Prewitt radius of the Mg^2+^ (72 pm) is between the values of octahedrally coordinated Ni^2+^ (69 pm) and Cu^2+^ (73 pm) ions [31]. Its hydration enthalpy (−1921 kJ/mol) is close to all of the investigated M(II) cations (for example, −1996 kJ/mol for Co^2+^ and −2105 kJ/mol for Ni^2+^ [32]). In addition, the water solubility of the Mg^2+^ salts is also not outstandingly different from those of the other salts. Therefore, the interpretation of the lowest position of the Mg^2+^ ions and, thus, the difficulties experienced during synthesis of the MgAl_4_–X^n^^−^–LDHs, is still not explained. It is conceivable that the answer is hiding in the water-exchange rate constants since the Mg^2+^ ion has a similar rate to that of the Co^2+^, which are largely lower compared to the values of Zn^2+^ and Cu^2+^ [33]. However, the Ni^2+^ cations show the slowest ligand exchange, which can be compensated for by its small ionic radius (the smallest among the cations studied). The mechanism of the entry of the metal ions into the vacant octahedral holes of the aluminum trihydroxide structure is still not fully clarified: the two competing theories are the solid-state topotactic imbibition of the metal cations and dissolution–reprecipitation [11,34]). On the basis of the data shown above, the mechanism must involve a step or steps where the metal cations partially or totally lose the hydrating water molecules, the rate of which affects the efficiency of the incorporation.

### 3.3. Optical Properties and Morphology of the Solids

The optical properties, the coordination and oxidation state, and the energy bandgap values of the materials prepared were determined using the UV-Vis DRS technique. The spectra of the solids contain a large number of bands. In all cases (Appendix A), around 230, 240 and 255 nm, the peaks were related to the oxygen-to-metal charge transfer transitions, and the band of the interlamellar nitrate anions was clearly observable at 300–310 nm [35]. As expected, for the MgAl_4_–, ZnAl_4_–LDHs and MgZnAl_8_–LTH, there were no other adsorptions in the investigated wavenumber range (Appendix A). However, the most crowded spectra were connected to the nickel-containing solids, due to the incorporation of the Ni^2+^, both with tetrahedral and octahedral coordination (Appendix A). The peaks at 365 and 415 nm can be ascribed to the d-d transition of the octahedral nickel cations, while the adsorption at ~645 nm showed the presence of nickel with tetrahedral geometry [36,37]. At around 745 nm for the copper-containing, 450 and 510 nm for the cobalt-containing solids, the spectra signified that these cations are octahedrally coordinated (Appendix A) [38]. In most of the cases, the corresponding wide peaks indicate their remarkably distorted environments.

It is worthwhile discussing the situation when the synthesis of the CoAl_4_–NO_3_^−^-LDH was not successful, using a 1:1 Co:Al initial molar ratio (Appendix A). In spite of the absence of LDH reflections, the incorporation of the Co^2+^ ions commenced and, thus, the solid that was prepared had a bluish color. As opposed to the adsorption profile of the CoAl_4_–NO_3_^−^–LDH obtained at a 4:1 initial molar ratio of Co:Al, in this case, the spectrum showed the presence of tetrahedrally coordinated Co^2+^ with a broad peak from 500 to 700 nm (maxima at 625 nm, Appendix A) [39]. Although in most cases, the incorporation of the octahedral metal cations occurred, there are some examples of tetrahedral geometry in these aluminum-rich LDHs [19,40]. In addition, in our recent study, tetrahedral Ni^2+^ coordination was seen in certain NiAl_4_ solids [24]. Therefore, it might be speculated that the formation of these types of Al(OH)_3_-based LDHs commences with the insertion of the metal cations into tetrahedral geometry.

Finally, in most of these cases, the evolution of the direct and indirect optical band gaps indicated a significant redshift in the energy values, compared to the starting milled gibbsite reagent (Table 1 and Appendix A). This shift was largest for the incorporation of Cu^2+^, and was highly dependent on the number of cations. Interestingly, the incorporation of the Mg^2+^ resulted in some blue-shift relative to the values of the corresponding solids.

The morphology of the LDHs when synthesized was largely amorphous and slightly laminated, resembling that of milled gibbsite [15], but disk-like particles were observable in several cases, like those of LiAl_2_–LDH [41]. These features are clearly seen in ZnAl_4_– and MgAl_4_–LDHs (200–300 nm diameter and 20–40 nm width) and, to a lesser extent, in CuAl_4_–LDH (300–400 nm diameter and < 100 nm width) with nitrate anions (Figure 2a). For the perchlorate-containing CuAl_4_–LDH and the nitrate-containing CoAl_4_–LDH, only the gibbsite-like morphology with significant deformation was registered (Figure 2b and Appendix A).

### 3.4. Catalytic Oxidation of Carbon Monoxide over the Aluminum-Rich LDHs as Catalysts

The imperfect combustion of fossil fuels results mainly in the formation of the highly toxic carbon monoxide (CO) from industrial as well as personal use (engines of vehicles, gas boilers). Therefore, recently, the catalytic oxidation of CO has received considerable attention. In recent years, the focus is increasingly on its oxidation at low temperatures (below 250–300 °C) over cost-effective catalysts (substituting the generally used noble metals) [42,43]. The Al-rich LDHs are promising precursors for these catalysts: they have great potential due to the atomic distribution of the low-transition metal content in the layers. Moreover, the high aluminum content could prevent sintering of the in situ formed particles in M(0) (elemental) form [44]. In addition, the heat-induced topotactic transformation of LDHs means the retaining of their ordered layered framework even after the removal of the molecules from the interlayer space, leaving behind highly porous residues.

In this research, the main goal was to map, firstly, the catalytic feasibility of these special M(II)-poor layered double hydroxides and to investigate the influence of the temperature rising on the conversion values, regarding the structural transformations and the departure of the interlamellar spaces. Therefore, the commonly applied pre-activation (oxidation, reduction) steps had to be skipped to get a clearer picture. Relative to the performance of the milled starting gibbsite reagents, the incorporation of the extraneous metal cations could increase the conversion of the CO, and products other than CO_2_ were not detected (Figure 3).

First, the M(II)Al_4_–NO_3_^−^–LDHs were tested as catalyst precursors. In the case of the zinc-containing sample, the oxidation commenced at 400 °C, while it had already begun at 300 °C for the nickel, cobalt and copper cation-containing ones. Interestingly, the MgAl_4_–NO_3_^−^–LDH precursor displayed the worst performance, but, with the exception of the milled Al(OH)_3_, all materials showed largely stable or enhancing CO transformation capabilities during the long-term activity tests. However, the cobalt- and copper-incorporated LDHs attested to higher conversion at the beginning of the process (at 300 °C); they showed some lags compared to the activity of the NiAl_4_–LDH between 350 and 550 °C. 

Therefore, the combination of the metal contents by preparing NiCuAl_8_– and NiCoAl_8_–LTHs (using the solids obtained during the syntheses with 0.20:1:1 Ni:Cu:Al and 1:21:1 Ni:Co:Al initial molar ratios) seemed like a logical step to intensify the CO transformation potential. No beneficial effect was observed in the case of the NiCuAl_8_–LTH; however, the incorporation of the cobalt cation next to the Ni(II) turned out to be useful. The CO conversion was found to increase up to ~60% from ~38% at a 300 °C reaction temperature, and CO_2_ formation was already observed at 250 °C (compared to the performance of the CoAl_4_–LDH). Finally, coke formation over the catalysts was not experienced on any occasion at the end of the long term activity tests (at 700 °C, Appendix A), and the XRD measurements detected only the reflections of the spinel-like phases (according to the crystal database card of the substoichiometric nickel aluminate, JCPDS#81-0710).

Commonly, LDHs have numerous well-separated weight losses above 100/150 °C and, thus, their application in pristine form was highly limited. All LDHs showed similar thermal behavior with endothermic processes; the incorporation of nickel [24], cobalt, copper, zinc and even magnesium ions did not result in significant variations (Appendix A). The departure of the physically adsorbed water occurred between 100 and 200 °C, and the interlayer water molecules were removed generally until 300 °C. At higher temperatures, the decomposition of the interlamellar nitrate anions and the dehydroxylation of the layers took place. However, the mass losses always showed some overlap (the departure of the surface hydroxide parts and physically adsorbed water occurred partially in parallel as the evaporation of the interlamellar H_2_O and the removal of the interlayered anions/structural hydroxides). The state of the catalysts was mainly the mixture of the metal oxides/hydroxides, with minimal adsorbed/intercalated water content in the temperature range of the commencement of CO oxidation (250–400 °C).

The CO chemisorption and the dehydration/dehydroxylation of the catalysts was followed by DRIFT measurements, applying 10% CO/He flow (for all samples, representative examples are shown for milled Al(OH)_3_ and NiCoAl_8_–LTH in Appendix A and for NiAl_4_–LDH in Figure 4). 

The gradual decrease of the structural and crystalline water content is indicated by the growth of the negative peaks around 3500 cm^−1^ and 1650 cm^−1^. While the double bands in the 2370–2320 cm^−1^ and in the 2180–2110 cm^−1^ regions show the presence of gaseous carbon dioxide and monoxide, respectively, the peaks under 1800 cm^−1^ wavenumber could be attributed to the different adsorbed carboxylate/carbonate/formate species formed by the reaction of the CO/CO_2_ molecules and surface Al/M(II)-OH groups [45]. Due to the high sensitivity of this technique for CO_2_, their formation was detectable at somewhat (50–100 °C) lower temperatures than by gas chromatography. Finally, a small shoulder (2050 cm^−1^) next to the vibration of gaseous CO molecules appeared at higher temperatures, which can be assigned to the adsorption of CO to metallic nickel [46,47], presumably generated during the redox reaction between the Ni(II) and CO. These data demonstrate that the CO chemisorption and, thus, the oxidation process is closely related to the elemental and ionic form of the metals in the LDH/LTH phases.

## 4. Conclusions

Based on gibbsite modification, a novel and facile synthesis route was further developed to prepare copper-poor and aluminum-rich CuAl_4_-type layered double hydroxides in phase-pure form for the first time. The chemical quality of the impregnating solution was varied, using copper bromide, chloride, nitrate and perchlorate starting salts. The formation of the expected copper hydroxide impurities could be minimized for the syntheses with bromide and chloride salts and totally eliminated when nitrate and perchlorate salts were used. The research also included the incorporation of Mg(II), Ni(II), Co(II) and Zn(II) ions and, thus, led to the preparation of layered triple and multiple hydroxides. Their composition yielded a preference order of the metals in the well-known affinity series of the incorporating cations. Compared to the Co(II), the imbibition of the Ni(II) and Cu(II) was about 21 and 9 times more favored, respectively, while this number was only around 1.5 in the case of competition with Zn(II). The last place was occupied by magnesium ions (possibly owing to their slow water-exchange rate).

The scanning electron microscopy images revealed disk-like particles. The optical analysis of the solids showed mainly the incorporation of the metal ions in octahedral form but, in a few cases, the tetrahedral coordination of the nickel and cobalt was also registered. The thermogravimetric analysis indicated thermal behavior independent from the quality of the incorporated metal ions. The cobalt-, nickel- and copper-containing materials offered enhanced carbon monoxide oxidation performances relative to those of the starting gibbsite, ZnAl_4_– and MgAl_4_–LDHs. The catalytic tests highlighted the remarkable catalytic potential of the layered hydroxides with mixed transition metal contents. 

## Figures and Tables

**Figure 1 materials-14-04880-f001:**
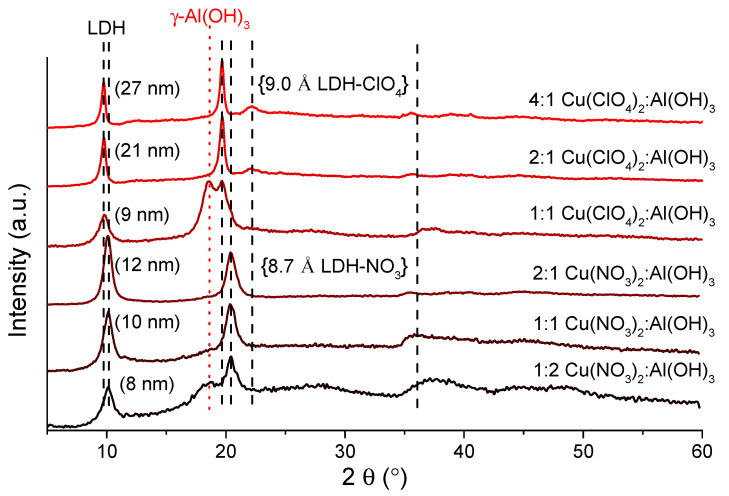
XRD patterns of the CuAl_4_–X^n^^−^–LDHs, formed with various X^n^^−^ interlayer anions. The initial Cu:Al molar ratio is shown to the right side of the XRD patterns; crystallite thicknesses are denoted next to the first reflection. Reaction conditions: 6 h pre-milling, 96 h stirring, 90 °C reaction temperature. The basal spacings in two representative cases are shown in curly brackets.

**Figure 2 materials-14-04880-f002:**
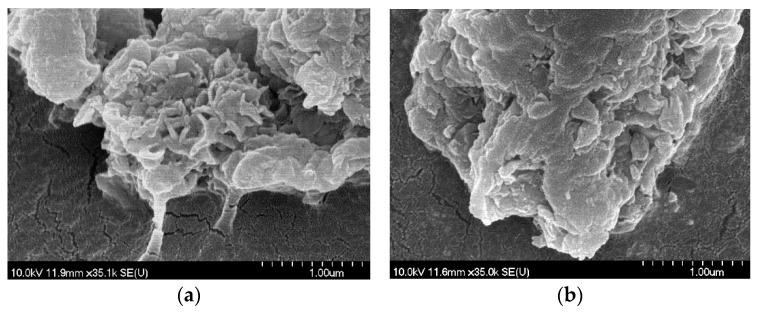
SEM photos of CuAl_4_–LDHs with nitrate (**a**) and perchlorate (**b**) interlayer anions.

**Figure 3 materials-14-04880-f003:**
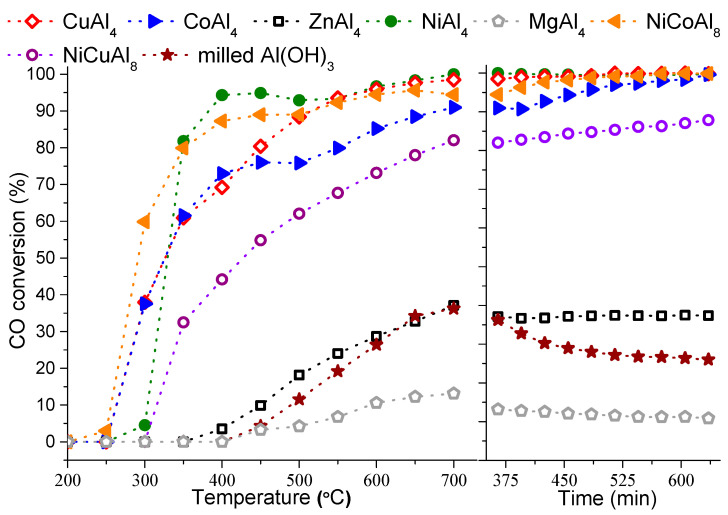
CO conversion values and long-term activity tests (at 700 °C) of the milled Al(OH)_3_, and the as-prepared LDHs and LTHs, at various reaction temperatures.

**Figure 4 materials-14-04880-f004:**
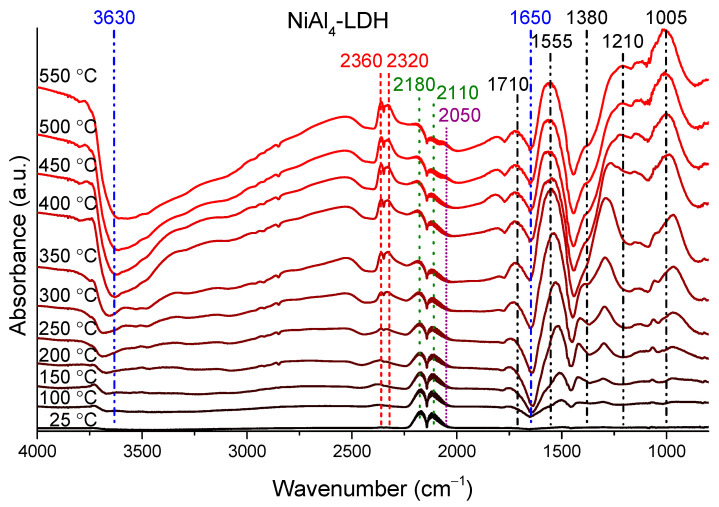
DRIFT spectra of the NiAl_4_–LDH, heated up to 550 °C in the presence of carbon monoxide–helium flow.

**Table 1 materials-14-04880-t001:** The molar ratios of the incorporated metal ions into the gibbsite structure; the initial and the measured values in the formed LTHs/LMHs, and their direct and indirect optical band gap values.

Samples (LDH-NO_3_)	Initial Molar Ratios ^1^					Measured Molar Ratios					Direct Band Gap (eV)	Indirect Band Gap (eV)
	Mg	Ni	Co	Cu	Zn	Mg	Ni	Co	Cu	Zn		
NiCu-Al	-	0.125	-	1	-	-	1	-	2.21	-	4.71	3.97
	-	0.17	-	1	-	-	1	-	1.33	-	4.77	4.08
	-	0.20	-	1	-	-	1	-	1.18	-	4.66	3.90
	-	0.25	-	1	-	-	1.35	-	1	-	4.72	4.05
	-	0.50	-	1	-	-	2.77	-	1	-	4.73	4.09
	-	1	-	1	-	-	5.05	-	1	-	4.93	4.19
	-	1	-	2	-	-	3.61	-	1	-	4.95	4.36
	-	1	-	4	-	-	1.85	-	1	-	4.96	4.49
NiCo-Al	-	1	1	-	-	-	21.24	1	-	-	4.96	4.37
NiZn-Al	-	1	-	-	1	-	16.70	-	-	1	5.01	4.60
CoZn-Al	-	-	1	-	1	-	-	1	-	1.48	5.10	4.76
CoCu-Al	-	-	1	1	-	-	-	1	8.76	-	4.56	3.78
CuZn-Al	-	-	-	4	4	-	-	-	7.01	1	4.62	3.87
NiCoCu-Al	-	1	1	1	-	-	22.11	1	6.15	-	4.97	4.39
NiCuZn-Al	-	1	-	1	1	-	15.80	-	3.73	1	4.95	4.39
NiCoZn-Al	-	1	1	-	1	-	21.84	1	-	1.33	4.92	4.49
CoCuZn-Al	-	-	1	1	1	-	-	1	11.24	1.45	4.80	4.06
NiCoCuZn-Al	-	1	1	1	1	-	17.67	1	5.23	1.44	4.95	4.46
MgNiCoCuZn-Al	1	1	1	1	1	0.03	17.03	1	5.31	1.36	5.03	4.61
MgCoCu-Al	1	-	1	1	-	0.06	-	1	7.81	-	4.87	4.21
MgCoZn-Al	4	-	4	-	4	0.11	-	1	-	1.32	5.21	4.91
MgCuZn-Al	1	-	-	1	1	0.06	-	-	6.15	1	4.83	4.20
MgCo-Al	4	-	4	-	-	1	-	22.47	-	-	5.00	4.34
MgCu-Al	1	-	-	1	-	1	-	-	122.2	-	4.88	4.07
MgZn-Al	4	-	-	-	4	1	-	-	-	45.63	5.14	4.83

^1^ The initial molar ratio of the aluminum was at 1 in every case.

## Data Availability

All the data is available within the manuscript and the electronic Appendix A.

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
