# Peer review of "M(II)Al4 Type Layered Double Hydroxides—Preparation Using Mechanochemical Route, Structural Characterization and Catalytic Application"

_materials, 2021, doi:10.3390/ma14174880_

Round 1
Reviewer 1 Report
The manuscript presents the synthesis, characterization and potential catalytic properties of high-Al layered double hydroxides. The main LDH was a CuAl4-based LDH. The incorporation of Mg2+,Ni2+, Co2+ and Zn2+ along Cu2+ were also studied and a preference order of the metals is established in the known affinity series of incorporating cations. A comparative study of catalytic oxidation of carbon monoxide of these M(II)Al4-LDHs shows the catalytic potential of LDHs with mixed transition metal contents.
The manuscript is well written, contains pertinent data and is suitable for publication considering the contribution to a less studied domain, of high amount of Al- LDHs.
My only requirement is regarding in more precise description of structural analysis results. The structural analysis is focusing only on the basal spacing and on the “crystal” size derived from the (003) reflection. This Scherrer crystallite size is measuring actually the coherently crystallographic domains along the c-axis, which is the axis of layers packing and this aspect should be described. “Crystal “suggests a very crystalline material: is more likely crystallite. What about the information regarding (110) reflection around 60.5 deg related to the “brucite-like” layer? I guess that probably this reflection does not appear clearly due the XRD patterns acquisition conditions: low power of the incident beam and high speed. Furthermore for the samples for which the (003) peaks has a small intensity while the (006) peak has a higher one, the “ crystal” size is not so clear define. What about the amorphous amount on each sample? Please add a comment regarding these aspects.
Author Response
Responses to the suggestions and comments of Reviewers
The manuscript has been corrected following the remarks and suggestions of the Reviewers. Changes made are written in red in the revised manuscript, and our detailed answers are given in the followings.
Reviewer #1
The manuscript is well written, contains pertinent data and is suitable for publication considering the contribution to a less studied domain, of high amount of Al- LDHs
- My only requirement is regarding in more precise description of structural analysis results. The structural analysis is focusing only on the basal spacing and on the “crystal” size derived from the (003) reflection. This Scherrer crystallite size is measuring actually the coherently crystallographic domains along the c-axis, which is the axis of layers packing and this aspect should be described.
Thank you very much, the requested information have been highlighted in the section of the powder X-ray diffractometry measurement description.
- “Crystal “suggests a very crystalline material: is more likely crystallite.
The authors thank for the advice, the crystal term has been replaced by the crystallite word in the relevant cases.
- What about the information regarding (110) reflection around 60.5 deg related to the “brucite-like” layer? I guess that probably this reflection does not appear clearly due the XRD patterns acquisition conditions: low power of the incident beam and high speed.
Indeed, the X-ray diffraction patterns of the solids up to higher 2 theta angles (up to 80° 2θ) did not reveal further reflections presumably due to the mentioned XRD measuring parameters, However it is worthy to note that the intensity of the (110) reflection is generally extremely low compared to that of the first and commonly most intense reflections in the case of the Al-rich LDHs [1-4].
- Williams, G.R.; Moorhous, S.J.; Prior, T.J.; Fogg, A.M.; Rees, N.H.; O’Hare, D. New insights into the intercalation chemistry of Al(OH)3. Dalton Trans. 2011, 40, 6012–6022.
- Chitrakar, R.; Makita, Y.; Sonoda, A.; Hirotsu, T. Synthesis of a novel layered double hydroxides [MgAl4(OH)12](Cl)22.4H2O and its anion-exchange properties. J. Hazard. Mater. 2011, 185, 1435–1439.
- Britto, S.; Kamath, P.V. Synthesis, structure refinement and chromate sorption characteristics of an Al-rich bayerite-based layered double hydroxide. Solid State Chem. 2014, 215, 206–210.
- Jensen, N.D.; Duong, N.T.; Bolanz, R.; Nishiyama, Y.; Rasmussen, C.A.; Gottlicher, J.; Steininger, R.; Prevot, V.; Nielsen, U.G. Synthesis and structural characterization of a pure ZnAl4(OH)12(SO4)·2.6H2O layered double hydroxide. Chem. 2019, 58, 6114–6122.
- Furthermore for the samples for which the (003) peaks has a small intensity while the (006) peak has a higher one, the “ crystal” size is not so clear define.
The Reviewer is right, in these instances, the calculation of the crystallite sizes based on the better evolved (006) reflections could give us more reliable data, however, the changes of the full width at half maximum values of the first and second reflections showed similar tendencies for every cases. It is worth to note that the signal of the unreacted aluminum hydroxide starting reagents is close to that of the (006) reflection around 20° 2θ and this could severely limit the fitting of the Gaussian curves and thus the calculation of the crystallite sizes for several times.
- What about the amorphous amount on each sample?
The molar ratio of the M(II):Al(III) metals were close to 1:4 in all cases (using the optimal synthesis parameters) indicated the presence of the amorphous gibbsite starting reagent only in minute amount. The calculation of the amount of amorphous form of the LDH phases was much more complicated due to the small differences observed in the physicochemical properties of the amorphous and crystalline parts. However, noticeable rise in the baseline stemmed from the disorders in the amorphous phases and/or and the generated fluorescence radiation during the XRD measurements for incorporation of Co and Ni metals was registered rarely, their amount estimated were varied in wide range between 60 and 4 wt% (using the Xpowder software package calculated from the integrated reflection intensities [5]) and showed clear correlation with the success of the LDH preparations. Applying the optimal parameters for the synthesis of the LDHs, LTHs and LMHs the amount of the amorphous particles could be decreased and remained generally under 4-7 w/w%. The corresponding data have been added to the section 3.1 of the manuscript.
- Hubbard, C.R.; Evans, E.H.; Smith, D.K. The reference intensity ratio, I/Ic, for computer simulated powder patterns. Appl. Crystallogr. 1976, 9, 169–174.
Reviewer 2 Report
The manuscript deals with the synthesis of a series of layered double hydroxides, layered triple hydroxides and layered multiple hydroxides rich in aluminum by utilizing mechanochemical treatment in order to activate gibbsite.
The paper is valuable in terms of novelty, but it is somehow difficult to read due to the multitude of synthesized and characterized samples, so I recommend publishing after a minor revision.
Some of my observations are presented below.
I understand that LTHs and LMHs have been synthesized from divalent cations of nitrates and perchlorates salts, using different ratios between the cations in the layered hydroxide structure. Perhaps, if for each synthesized hydroxide formula the diffractograms of the materials with the most favorable cation ratios were presented in a figure or two, it would be better to observe the obtaining of pure phases and stratified structures in the synthesized samples. It is difficult to trace, for each individual hydroxide formulation, which of the ratios of cations used is the optimal ratio.
On page 6, rows 263-264, specify: “… for the synthesis of the CoZnAl-, MgCoAl- and MgZnAl-LTHs and the MgCoZnAl-LMH, 4:4:1 and 4:4:4:1 initial molar ratios were employed, respectively, to prepare phase-pure LTH and LMH systems.” Nowhere in the manuscript (tables or figures) did I find data on triple or multiple hydroxides with these molar ratios. What would be the explanation?
Author Response
Responses to the suggestions and comments of Reviewers
The manuscript has been corrected following the remarks and suggestions of the Reviewers. Changes made are written in red in the revised manuscript, and our detailed answers are given in the followings.
Reviewer #2
The paper is valuable in terms of novelty, but it is somehow difficult to read due to the multitude of synthesized and characterized samples, so I recommend publishing after a minor revision. Some of my observations are presented below.
- I understand that LTHs and LMHs have been synthesized from divalent cations of nitrates and perchlorates salts, using different ratios between the cations in the layered hydroxide structure. Perhaps, if for each synthesized hydroxide formula the diffractograms of the materials with the most favorable cation ratios were presented in a figure or two, it would be better to observe the obtaining of pure phases and stratified structures in the synthesized samples. It is difficult to trace, for each individual hydroxide formulation, which of the ratios of cations used is the optimal ratio.
The authors thanks the advice, indeed the numerous solids prepared and their XRD patterns made it a bit difficult to follow the process of the syntheses toward to get the optimal parameters to reach the phase-pure preparations. In the case of the copper containing LDHs with chloride, bromide, nitrate and perchlorate interlayer anions, our goal was to find the necessary copper salts and gibbsite starting reagent molar ratios for the inhibition of the formation of various side products, therefore the corresponding XRD patterns were displayed together to follow easily the diffractograms. However, for the preparation of layered triple and multiple hydroxides, the formation of any side products were less observable due to the previously determined and applied optimal reaction conditions, therefore our attention was focused on the determination of the built in metal molar ratios in order to establish and expand the well-known selectivity series of the incorporating cations. Therefore due to the large number of the prepared LTHs and LMHs, only a few relevant XRD curves have been inserted to the supplementary information text to demonstrate their similar pattern compared to the LDH variants. Finally, in the synthesis of the MgAl4-LDHs, our aim was only to show the necessity of the extremely high Mg:Al molar ratio and long premilling duration to achieve Mg incorporation. However, this was more or less expected on the basis of the literature data.
- On page 6, rows 263-264, specify: “… for the synthesis of the CoZnAl-, MgCoAl- and MgZnAl-LTHs and the MgCoZnAl-LMH, 4:4:1 and 4:4:4:1 initial molar ratios were employed, respectively, to prepare phase-pure LTH and LMH systems.” Nowhere in the manuscript (tables or figures) did I find data on triple or multiple hydroxides with these molar ratios. What would be the explanation?
The XRD patterns of the CoZn-, MgCo-, MgZn- and MgCoZn-Al solids were not presented in the manuscript or the supporting information, partly due to the large number of the XRD patterns recorded and partly to the fact that the patterns did not reveal any specific information different from that of the NiCuAl-LTHs. Only the XRD patterns of the CoAl- and ZnAl-LDHs were displayed, with 1:1 and 4:1 cobalt and zinc nitrate and gibbsite starting molar ratios. In this figure, it is clearly observable that the low cobalt and zinc amount resulted in only the formation of the boehmite, the dehydrated form of the gibbsite, therefore the successful synthesises of the above mentioned LTHs and LMH were expected to be achieved using the high excess of the M(II) cations. Furthermore, the UV–Vis–DR spectra of the solids shown in figure S11 and S12 verify/indicate the presence of the nitrate anions and the tetrahedrally coordinated Co2+ cations in remarkably distorted environments. The corresponding data have been added to the section 3.2 and the Table 1 of the manuscript.